# Ketogenic Diet: Impact on Cellular Lipids in Hippocampal Murine Neurons

**DOI:** 10.3390/nu12123870

**Published:** 2020-12-18

**Authors:** Partha Dabke, Graham Brogden, Hassan Y. Naim, Anibh M. Das

**Affiliations:** 1Clinic for Pediatric Kidney, Liver, and Metabolic Diseases, Hannover Medical School, Carl-Neuberg Str. 1, 30625 Hannover, Germany; dabke.partha@mh-hannover.de; 2Institute for Physiological Chemistry, University of Veterinary Medicine, Bünteweg 17, 30559 Hannover, Germany; graham.brogden@tiho-hannover.de (G.B.); hassan.naim@tiho-hannover.de (H.Y.N.)

**Keywords:** ketogenic diet, neurons, cholesterol, phospholipids, hydroxybutyrate, decanoic acid

## Abstract

Background: The mechanism of action of the ketogenic diet (KD), an effective treatment for pharmacotherapy refractory epilepsy, is not fully elucidated. The present study examined the effects of two metabolites accumulating under KD—beta-hydroxybutyrate (ßHB) and decanoic acid (C10) in hippocampal murine (HT22) neurons. Methods: A mouse HT22 hippocampal neuronal cell line was used in the present study. Cellular lipids were analyzed in cell cultures incubated with high (standard) versus low glucose supplemented with ßHB or C10. Cellular cholesterol was analyzed using HPLC, while phospholipids and sphingomyelin (SM) were analyzed using HPTLC. Results: HT22 cells showed higher cholesterol, but lower SM levels in the low glucose group without supplements as compared to the high glucose groups. While cellular cholesterol was reduced in both ßHB- and C10-incubated cells, phospholipids were significantly higher in C10-incubated neurons. Ratios of individual phospholipids to cholesterol were significantly higher in ßHB- and C10-incubated neurons as compared to controls. Conclusion: Changes in the ratios of individual phospholipids to cholesterol in HT22 neurons suggest a possible alteration in the composition of the plasma membrane and organelle membranes, which may provide insight into the working mechanism of KD metabolites ßHB and C10.

## 1. Introduction

The brain is one of the most lipid-rich organs with lipids constituting about 50% of its dry weight [1]. Lipids of different classes have diverse subcellular localizations, mainly linked to their functions: cholesterol and sphingomyelin are mainly localized to the plasma membrane, whereas phospholipids are present in high amounts in the endoplasmic reticulum (ER) and the mitochondria [2]. CNS lipids are essential structural components not only of the cellular plasma membrane, but also of membranes in various cellular organelles [3]. Phosphatidylcholine (PC), phosphatidylethanolamine (PE), phosphatidylserine (PS), and cardiolipin (CL) play an important role in forming and maintaining the structures of biomembranes [2]. PC, apart from being a major component of the plasma membrane, also provides choline to neurons for the synthesis of acetylcholine as a neurotransmitter [4]. PE and PS are both predominantly present on the inner side of the plasma membrane [5]. Additionally, PE is also found to be an important component of the inner mitochondrial membrane [6]. CL, a major constituent of the inner mitochondrial membrane, plays an important role in the maintenance of mitochondrial membrane potential and oxidative phosphorylation (OXPHOS) by regulating several mitochondrial proteins and the proton leak (“uncoupling”) during respiration [7,8]. Phospholipids, sphingolipids, and cholesterol are instrumental in the formation of lipid rafts [9], which are important components in cellular signaling and transport [10]. Ketone bodies formed by fatty acid oxidation in the liver are important energy substrates for the brain, especially during fasting conditions or under a ketogenic diet (KD) treatment [11]. For a detailed review of lipids as energy substrates, see Tracey et al. [2]. Cholesterol in the brain accounts for about 25% of the overall amount in humans [12] and about 15% in mice [13]. Cholesterol is a key component of myelin and thus, extremely high concentrations of cholesterol can be found in the CNS of mammals—owing to the presence of many myelinated nerve fibers for signal transmission [14]. In the brain, cholesterol is predominantly endogenously produced and its concentration does not depend on variations of the circulated cholesterol [15,16].

The ketogenic diet (KD) is a high-fat and low-carbohydrate diet, various forms of which have been used for the treatment of pharmaco-refractory epilepsy [17]. Apart from epilepsy, the KD has been used in the treatment of neurodegenerative disorders [18], inborn errors of metabolism (IEM) [19,20,21], and cancer [22]. The classical KD was developed in the early 20th century (refer to [23] for a detailed history of KD development). The MCT diet, consisting of medium chain fatty acids—predominantly decanoic acid (C10) and octanoic acid (C8)—was shown to be as effective as the classical KD in controlling seizures in children with therapy refractory epilepsy [24,25]. However, despite several proposed hypotheses, it is still not completely clear as to how the KD exerts its therapeutic effect [26].

Aim of the study: The current study aimed at investigating the effects of caloric restriction and two important KD metabolites—beta-hydroxybutyrate and decanoic acid—on cellular lipid composition in HT22 hippocampal murine neurons in vitro. Cholesterol and phospholipid content was measured under the conditions mentioned above.

## 2. Materials and Methods

Cell culture: The HT22 murine hippocampal neuronal cell line was used as a cell model for this study. Cells were cultured in T75 culture flasks (Sarstedt AG, Nümbrecht, Germany) using Dulbecco’s Modified Eagle Medium (DMEM) (Thermo Scientific, Waltham, MA, USA) with 10% (*v*/*v*) fetal bovine serum (Biowest SAS, Nuaillé, France) and 1% (*v*/*v*) Penicillin/Streptomycin (Sigma-Aldrich, Darmstadt, Germany). The cells were cultured at 37 °C and 5% CO_2_. The standard glucose concentration in DMEM used to culture HT22 cells is 4.5 g/L (22.5 mmol/L). In order to simulate a “ketogenic” environment in vitro, we established HT22 cell cultures in low glucose, i.e., 1 g/L (5.5 mmol/L) glucose concentration in DMEM which was the lowest concentration tolerated in terms of cell growth. The cells were incubated without supplements (control) or in the presence of 5 mM ßHB or 250 µM decanoid acid (C10), respectively, to mimic a KD environment for one week; the medium was changed every second day. While 5 mM is the approximate plasma concentration of ßHB found in patients undergoing KD treatment, the C10 concentration (250 µM) was adopted from blood C10 concentrations in patients [27] and previous work with C10 and neuronal cells [28]. CNS levels of these metabolites are slightly lower compared to plasma levels [29].

Lipid isolation and quantification: Lipids were isolated using a methanol-, chloroform-, and water-based method as described previously [30,31]. After isolation, the whole sample was dissolved in 250 µL of chloroform and methanol solution (1:1). Cholesterol and phospholipid analyses were performed by HPLC and HPTLC, respectively. Cholesterol was quantified as described previously [32]. Briefly, a Hitachi Chromaster HPLC system fitted with a Chromolith^®^ HighResolution RP-18 end capped 100–4.6 mm column coupled to a 5–4.6 mm guard cartridge was used. Methanol was used as a mobile phase with a flow rate of 1 mL/min at 22 bar. The UV detector measured at 202 nm and samples were quantified against an external standard. Phospholipids were quantified by HPTLC as previously described [32]. Samples were spotted onto silica gel 60 plates (Sigma-Aldrich, St. Louis, MO, USA) and separated using three running solvents. After drying, lipids were stained using 7.5% phosphoric acid followed by charring at 170 °C for 10 min. The plates were then scanned and analyzed by CP ATLAS software (Lazarsoftware). Samples were identified against a standard run on the same plate.

Statistical analysis: The statistical analysis was performed using the GraphPad Prism 8 software (GraphPad Prism, San Diego, CA, USA). The unpaired t-test was used to calculate the differences between the low versus high glucose concentration groups and control versus ßHB/C10-treated groups. A *p*-value of <0.05 was considered as significant. All data are presented as mean + SD and a minimum of 3 independent samples were used in each experiment.

## 3. Results

### 3.1. Caloric Restriction Modulates Cellular Lipid Composition

Cholesterol and phospholipids were measured in the HT22 neuronal cells grown under two different glucose concentrations: high (4.5 g/L) and low (1 g/L) (Figure 1). A significant increase in cellular cholesterol in the low glucose group was observed. Amongst the phospholipids, only sphingomyelin levels were significantly decreased in the low glucose group. Cellular quantities of other phospholipids were not significantly different between the two groups.

### 3.2. Incubation with Beta-Hydroxybutyrate (ßHB) Leads to a Decrease in Cholesterol and Phosphatidylserine

HT22 neuronal cells were incubated for 1 week in DMEM with low glucose concentration (0.1%, wt/vol) in two groups—the control (untreated) and 5 mM ßHB-treated groups. Cellular cholesterol was significantly reduced in the ßHB-treated cells (Figure 2A). Amongst the phospholipids, only phosphatidylserine (PS) levels (Figure 2B) were significantly decreased in the cells incubated with ßHB. It was not clear why levels of other phospholipids remained unchanged in the ßHB-treated cells.

Ratios of individual phospholipids to cholesterol in both the control (untreated) and ßHB-treated groups were calculated. The individual phospholipid to cholesterol ratios (except that of PS to cholesterol) were 2- to 3-fold higher in the ßHB-treated group, compared to the untreated group. Cholesterol and phospholipids are largely localized in the plasma membrane as well as the membranes of cellular organelles [3]. An alteration in ratios of phospholipids to cholesterol may point towards a change in the composition of the various organelle and/or plasma membranes.

### 3.3. Decanoic Acid (C10) Elevates the Phospholipid and Sphingomyelin Levels

HT22 cells were incubated with decanoic acid (C10) for one week. A slight decrease in cholesterol levels was observed; however, this effect was not significant (Figure 3A). By contrast, the phospholipids and sphingomyelin quantities were significantly elevated (Figure 3B).

Apart from a decrease in the cholesterol levels, the results of the C10-incubation were opposite to that of the incubation with ßHB. The ratios of individual phospholipids and sphingomyelin to cholesterol, however, showed a similar trend in C10- and ßHB-incubated cells.

## 4. Discussion

KD treatment prominently alters the uptake of substrates by the brain. Ketone bodies are taken up as fuel by the brain and converted to acetyl-CoA-molecules which enter the TCA cycle in the mitochondrial matrix for energy production [11]. Concentrations of medium-chain fatty acids such as decanoic acid (C10) are elevated [27] along with ßHB. Although the mechanism is not yet clear, it has been shown that C10 can cross the blood–brain barrier [29]. Several mechanisms have been proposed via which the two KD metabolites, C10 and ßHB, may exert their anti-epileptic and neuroprotective effects [17,26,33,34]. Taha et al. proposed that polyunsaturated fatty acids (PUFAs) may contribute to the anti-seizure effect of KD [35]. This study showed that although ketosis was not sustained beyond one week in the animals fed with a high fat KD, there were marked changes in lipid metabolism. While levels of plasma PUFAs were reduced, a 15% increase in brain PUFAs was observed [35].

It was recently shown that levels of phospholipids and cholesterol in the brain tissue of rats with post traumatic epilepsy were altered [36]. Acharya et al. showed that picrotoxin-induced convulsions in rats led to alterations in lipid levels in the brain tissue along with structural changes in the microsomal membranes [37]. The study showed that while total brain cholesterol levels were elevated, levels of some phospholipids such as phosphoinositol (PI), PE, and PS were significantly reduced [37]. Our study showed that both the key metabolites of KD—ßHB and C10—decreased total cellular cholesterol; this may redress the elevated cholesterol levels observed in epilepsy [37]. HT22 neurons incubated with C10 showed significantly higher levels of PE and PS. However, incubation with ßHB resulted in a decrease in PS levels, while levels of other phospholipids remained almost unchanged. It is possible that simultaneous elevation of ßHB and C10 as occurs under KD may result in reduced cholesterol levels and elevated PE concentrations, while PS levels remain unchanged.

Another study in rats with picrotoxin-induced seizures demonstrated that the fluidity of the mitochondrial membrane in brain tissue was increased along with a significant decrease in cholesterol, PE, and PS levels and an increase in levels of PC and SM [38]. Higher ratios of cellular phospholipids to cholesterol were observed in the HT22 cells treated with C10 as well as ßHB in our study. Plasma membrane and the membranes of various subcellular organelles (including the mitochondria) contain cholesterol, phospholipids, and SM in varying proportions, determining the structure and function of these membranes [3,14]. It is not clear how alterations in the lipid composition of membranes translate into an anticonvulsive effect. Significant alterations of lipids (including ratios of phospholipids and SM to cholesterol) under KD treatment may lead to the altered physicochemical properties of biomembranes and play a role in the therapeutic effect of KD. Biomembranes, namely the plasma membrane of neurons, may be regarded as dielectric material of an electric capacitor separating the extra- and intracellular electrical charges. Alterations in lipid composition may have an impact on the fluidity, dielectric constant, and thickness of the plasma membrane, thus changing the membrane potential, and hence, excitability of the neurons. Furthermore, changes in lipid composition may impact membrane trafficking via transporters and cell signaling; however, this needs to be investigated in future studies, both in vitro and in vivo.

We recently demonstrated that sirtuins (a group of NAD+-dependent enzymes) and the mitochondrial respiratory chain complexes were upregulated in HT22 neurons under C10- and ßHB-incubation [39]. Lipids serve as energy substrates in the CNS (reviewed in [2]) and phospholipids such as PE and CL are crucial for the efficient functioning of the respiratory chain and maintaining the mitochondrial membrane potential [8,40]. Thus, an alteration in levels of PE and CL along with higher PE:Cholesterol and CL:Cholesterol ratios in C10- and ßHB-treated groups may mediate improved mitochondrial function, possibly via altered membrane structure, geometry, and function (e.g., dielectric constant as mentioned above). Upregulation of mitochondrial function has been proposed as an important mechanism of action of KD [17,28,41,42,43]. As mitochondrial dysfunction has been linked to neurodegenerative diseases, this mechanism may also explain the beneficial effect of KD not only on epilepsy, but on neurodegeneration as well.

The role of neuronal phospholipids in mediating the therapeutic effect of KD needs to be further investigated. In future studies, the effect of coincubating HT22 neurons with both C10 and ßHB on the lipid composition of membranes should be studied. Our study is based on a neuronal cell model and further in vivo experimentation is necessary to evaluate the potential effects of metabolites of KD on neuronal lipids. The effects of other KD metabolites on cellular lipid composition will be studied in the future in neuronal as well as other cells like hepatocytes.

Our study has limitations. We only evaluated two metabolites which are elevated under KD and did not perform an in vivo study. Lipid composition should be studied at the subcellular level in organelles, namely mitochondria. Furthermore, the mechanism leading to altered lipid composition in biomembranes needs to be elucidated.

## 5. Conclusions

Significant alterations in cellular cholesterol, phospholipids, and sphingomyelin were observed in HT22 neurons under 250 µM C10 and 5 mM ßHB incubations in vitro (see Figure 4). These changes may be linked to the antiepileptic effect of KD as well as the protective effect of KD in neurodegenerative disorders.

## Figures and Tables

**Figure 1 nutrients-12-03870-f001:**
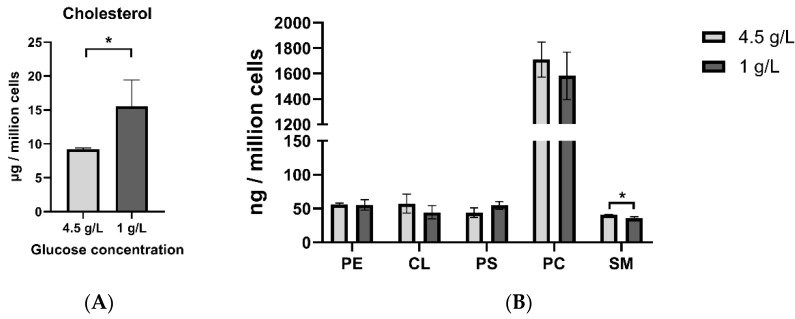
Cellular cholesterol (µg/million cells) (**A**) and phospholipid (ng/million cells) quantities (**B**) in HT22 neurons. * *p* < 0.05 (*n* = 3–6); data are expressed as mean + SD. CL—cardiolipin; PC—phosphatidylcholine; PE—phosphatidylethanolamine; PS—phosphatidylserine; SM—sphingomyelin.

**Figure 2 nutrients-12-03870-f002:**
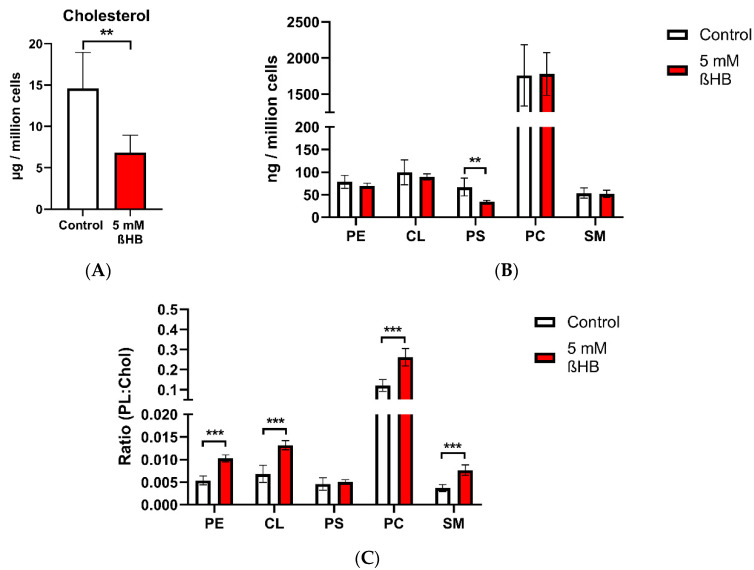
Cellular lipid quantities in HT22 neurons (control and ßHB-treated). Graph (**A**) represents cholesterol quantity (in µg per one million cells) measured using HPLC; graph (**B**) represents cellular phospholipid quantities (in ng per one million cells), measured using HPTLC; graph (**C**) represents the ratios of individual phospholipids to cholesterol in the control and beta-hydroxybutyrate-treated groups. Data are presented as mean + SD (*n* = 4–6); ** *p* < 0.01, *** *p* < 0.001. Chol—cholesterol; CL—cardiolipin; PC—phosphatidylcholine; PE—phosphatidylethanolamine; PL—phospholipid; PS—phosphatidylserine; SM—sphingomyelin.

**Figure 3 nutrients-12-03870-f003:**
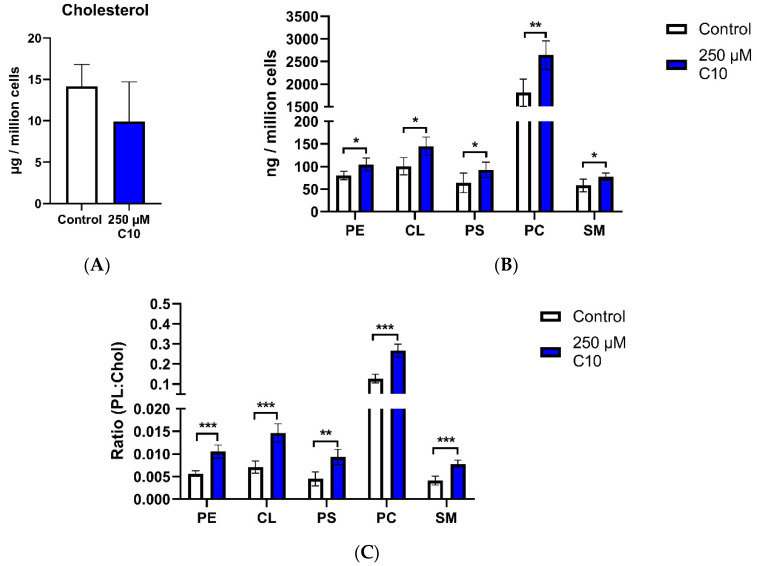
Cellular lipid quantities in HT22 neurons (control and C10 treated). Graph (**A**) presents cholesterol quantity (in µg per one million cells) measured using HPLC; graph (**B**) presents cellular phospholipid quantities (in ng per one million cells), measured using HPTLC; graph (**C**) presents ratios of individual phospholipids to cholesterol in the control and beta-hydroxybutyrate-treated groups. Data are presented as mean + SD (*n* = 4–6); * *p* < 0.05, ** *p* < 0.01, *** *p* < 0.001. Chol—cholesterol; CL—cardiolipin; PC—phosphatidylcholine; PE—phosphatidylethanolamine; PL—phospholipid; PS—phosphatidylserine; SM—sphingomyelin.

**Figure 4 nutrients-12-03870-f004:**
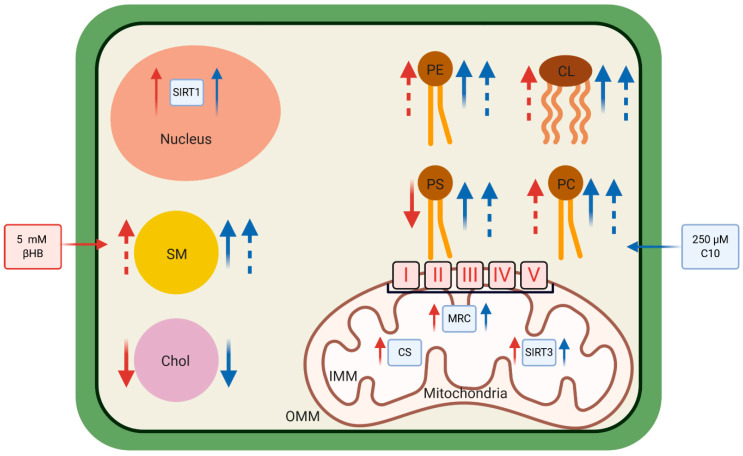
ßHB—Beta-hydroxybutyrate; C10—Decanoic acid; Cho—Cholesterol; CL—Cardiolipin; CS—Citrate Synthase; IMM—Inner mitochondrial membrane; MRC—Mitochondrial Respiratory Chain (I—V are the individual complexes of the MRC); OMM—Outer mitochondrial membrane; PC—Phosphatidylcholine; PE—Phosphatidylethanolamine; PS—Phosphatidylserine; SIRT1—Sirtuin 1; SIRT3—Sirtuin 3; SM—Sphingomyelin. All red arrows depict the effects of ßHB, and all blue arrows depict the effects of C10. Thin arrows show already published data (adapted from [39]). Solid thick arrows show the effects of ßHB and C10 on absolute quantities of the depicted cellular lipids. Dotted thick arrows represent the effects of ßHB and C10 on ratios of the individual phospholipids (and sphingomyelin) to cellular cholesterol. Figure created with BioRender.com.

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
