# Peer review of "Ketogenic Diet: Impact on Cellular Lipids in Hippocampal Murine Neurons"

_nutrients, 2020, doi:10.3390/nu12123870_

Round 1

Reviewer 1 Report

The paper is well written and of interest since the mechanisms through which KD exerts its effects are still under debate. As correctly recognized by authors, in vivo studies and in deep characterization of sub-cellular biochemical changes with KD regimen should be the final objective, in the mean time this paper reminds the importance of conducting further research in this field and reiterates the importance of targeting mitochondria. On this purpose, I would suggest to cite the paper of Kim et al,  'Ketone bodies mediate antiseizure effects through mitochondrial permeability transition', 2015 Ann Neurol when mentioning mitochondrial function and KD action in the discussion. 

It could be interesting to eventually investigate in vitro whether age could influence the changes in lipid composition described.

Finally, the paper might gain further interest if authors add some hypotheses about possible explanation of the link between changes observed and anti-convulsant effects.

Author Response

Dear Reviewer,

Thank you for reviewing our manuscript and your suggestions.

The reference of Kim et al has been added in the discussion related to mitochondrial function and effect of KD.

The discussion has been expanded and more hypotheses added as suggested.

Thank you

Kind Regards

Dr. Partha Dabke

Reviewer 2 Report

Here the authors describe the possible role of two metabolites of Ketogenic Diet (KD), beta-hydroxybutyrate (βHB) and decanoic acid (C10) in murine hippocampal neurons (HT22). Cells were treated with these two metabolites under standard or low glucose conditions, to simulate the ketogenic diet effects. They showed a marked reduction of cholesterol in the βHB treated cells, accompanied by an increase in phospholipids/cholesterol ratio. The same trend in cholesterol was found also in the C10-treated cell, although it was not significant. Nevertheless, a similar phospholipids/cholesterol ratio was found also in C10-treated cells. This condition can explain the beneficial effects of the ketogenic diet in epilepsy.

In my opinion, the work here presented is very well conducted and undoubtedly the authors showed very interesting data. The abstract is very clear and proceeds in a completely logical manner. Introduction and aims are well described, and methods are clearly presented. The results are shown in detail and the discussion is very well argued with specific references.

I have just a curiosity: why the authors have not tried to do a combined treatment of βHB + C10 to evaluate the effect of both metabolites at the same time? Is there any contraindication in carrying out the combined treatment? Did the authors plan to carry out this experiment in the future, perhaps on more cell lines? I realize that asking to do it now is a huge effort and investment of energy and material, but I think the authors might think about it for future works.

So I suggest only a minor revision and correction of some small oversight as in line 59 where the possessive adjective "its" is erroneously written "is".

Author Response

Dear Reviewer,

Thank you for reviewing our manuscript and your suggestions.

The effects of combined treatment of ßHB + C10 will be explored in the future, as rightly suggested by you. We have added this point in the discussion.

We have made the grammatical/spelling change in line 59. 

Thank you

Kind Regards

Dr. Partha Dabke

Reviewer 3 Report

I reviewed the article, "Ketogenic Diet: Impact on Cellular Lipids in 3 Hippocampal Murine Neurons", and found it to be of high interest.  Mechanistic papers regarding the therapeutic nature of the ketogenic diet are highly needed.  This paper was well designed and written.  

Author Response

Dear Reviewer,

Thank you for reviewing our manuscript and your comments.

Thank you

Kind Regards

Dr. Partha Dabke